# Gene co-expression network analysis of the human gut commensal bacterium *Faecalibacterium prausnitzii* in R-Shiny

**Sandrine Auger**[1]*, **Virginie Mournetas**[2]*, **Hélène Chiapello**[3], **Valentin Loux**[3,4], **Philippe Langella**[1], **Jean-Marc Chatel**[1]

**1** Université Paris-Saclay, INRAE, AgroParisTech, Micalis Institute, Jouy-en-Josas, France, **2** ADLIN Science, Pépinière « Genopole Entreprises », Evry, France, **3** Université Paris-Saclay, INRAE, MaIAGE, Jouy-en-Josas, France, **4** Université Paris-Saclay, INRAE, BioinfOmics, MIGALE Bioinformatics Facility, Jouy-en-Josas, France

* sandrine.auger@inrae.fr (SA); contact@virginie-mournetas.fr (VM)

## Abstract

*Faecalibacterium prausnitzii* is abundant in the healthy human intestinal microbiota, and the absence or scarcity of this bacterium has been linked with inflammatory diseases and metabolic disorders. *F. prausnitzii* thus shows promise as a next-generation probiotic for use in restoring the balance of the gut microbial flora and, due to its strong anti-inflammatory properties, for the treatment of certain pathological conditions. However, very little information is available about gene function and regulation in this species. Here, we utilized a systems biology approach—weighted gene co-expression network analysis (WGCNA)–to analyze gene expression in three publicly available RNAseq datasets from *F. prausnitzii* strain A2-165, all obtained in different laboratory conditions. The co-expression network was then subdivided into 24 co-expression gene modules. A subsequent enrichment analysis revealed that these modules are associated with different kinds of biological processes, such as arginine, histidine, cobalamin, or fatty acid metabolism as well as bacteriophage function, molecular chaperones, stress response, or SOS response. Some genes appeared to be associated with mechanisms of protection against oxidative stress and could be essential for *F. prausnitzii*'s adaptation and survival under anaerobic laboratory conditions. Hub and bottleneck genes were identified by analyses of intramodular connectivity and betweenness, respectively; this highlighted the high connectivity of genes located on mobile genetic elements, which could promote the genetic evolution of *F. prausnitzii* within its ecological niche. This study provides the first exploration of the complex regulatory networks in *F. prausnitzii*, and all of the "omics" data are available online for exploration through a graphical interface at https://shiny.migale.inrae.fr/app/faeprau.

## Introduction

The human gut microbiota plays a fundamental role in human health. The collective genomes of bacteria and other microorganisms in this ecosystem, including fungi, viruses, and archaea,

**Data Availability Statement:** We provide repository information for our data: the Gitlab repository (https://gitlab.com/adlin-science-public/FaePrau) and the shinyapp (https://faeprau.omics.

ovh/). The shinyapp is now freely accessible to the public via the URL https://shiny.migale.inrae.fr/app/faeprau.

**Funding:** This work was funded by the Institut National de la Recherche Agronomique, Alimentaire et de l'Environnement. The funders had no role in study design, data collection and analysis, decision to publish, or preparation of the manuscript.

**Competing interests:** The authors have declared that no competing interests exist.

have received increasing attention in the past two decades (Integrative HMP (iHMP) Consortium, 2019). Taken together, this research has revealed the extent to which the gut microbiome actively affects numerous host functions, including immune system development, maintenance of intestinal mucosal integrity, human metabolism, brain processes, and behavior [1–4]. The community composition and behavior of this assemblage can be influenced by a variety of factors, such as diet and lifestyle. Alterations to the gut microbiota, referred to as dysbiosis, can disrupt essential health-promoting services and are associated with a variety of illnesses, including Inflammatory Bowel Diseases (IBDs) and cancer [5–7]. Within the gut microbiota, microorganisms interact in multiple contexts and exhibit heterogeneous behaviors. The diversity and number of these interactions can generate unpredictable population dynamic, and the emergent properties and cross-scale interactions within these complex systems are best studied using comprehensive investigations of the entire community. In this context, the integrative approaches used in systems biology represent important tools for studying the interplay between the different biological systems within the whole, and may help to elucidate host microbiome interactions [8,9].

Among the strictly anaerobic bacteria present in the human colon, *Faecalibacterium praunsitzii* is one of the most common taxa and serves as a general health biomarker in humans [10]. The abundance of *F. prausnitzii* is reflective of the health status and colonic environment of the host. Low levels of this species in the gut are associated with various gastrointestinal disorders, such as IBDs, Irritable Bowel Syndrome, colorectal cancer, or obesity [11,12]. One of the mechanisms by which *F. prausnitzii* exerts a beneficial effects is the production of butyrate, which is involved in maintaining the gut lining and in fighting inflammation [13–15]. Moreover, this species produces several bioactive molecules that affect inflammation and gut barrier function, such as the microbial anti-inflammatory molecule (MAM) [16–18]. *F. prausnitzii* is thus considered a promising next generation probiotic (NGP) which can help to not only restore the balance of the microbial flora but also to aid in the treatment of certain pathological conditions [19,20]. Despite the importance of *F. prausnitzii* within the gut, nothing is known about the regulatory factors responsible for a range of important processes, including intestinal colonization, quorum sensing, and stress responses to bile salt, acidic pH, or oxidative stress. In this bacterium and for NGPs in general, the first step in understanding the interactions and behavior within the gut is to consider individual bacteria as a biological system, one that responds to environmental perturbations through complex networks of gene interactions.

Recent studies have utilized high-throughput RNA sequencing (RNA-seq) to examine gene expression in *F. prausnitzii* strain A2-165, mostly with the goal of identifying differential gene expression under different conditions [21–23]. However, certain underlying properties can only be explained by studying organisms as complex systems [24–26]. To this end, expression data can be used to accurately group genes into functional modules based on co-expression patterns. In particular, the algorithm developed for weighted gene co-expression network analysis (WGCNA) enables the construction of gene networks through consideration of the co-expression patterns between two genes as well as the overlap between neighboring genes [27]. Highly correlated genes are clustered into larger modules based on similarities in their expression profiles, and members of a given module are often involved in similar functional processes [28]. To date, WGCNA has been successfully used to construct gene co-expression networks in several bacteria, including *Mycobacterium tuberculosis* [29], *Escherichia coli* [30], *Lactococcus lactis* [31], *Vibrio cholerae* [32], and *Streptococcus oralis* [33].

In the current study, gene expression data were obtained for *F. prausnitzii* A2-165 from three public RNAseq datasets. Co-expression networks were analyzed using WGCNA and genes were clustered into modules with similar expression patterns. For each module, we investigated enrichment in biological functions. The structure of the network was also analyzed to identify

hub and bottleneck genes; of the 25 we identified, the majority were located in the region of six mobile genetic elements that could promote intracellular or intercellular DNA mobility. This study represents the first time that a gene expression network has been constructed for *F. prausnitzii*. In addition, an R-Shiny application is available online to make the data and the analyses accessible and usable to a wide audience (https://shiny.migale.inrae.fr/app/faeprau).

## Materials and methods

### Data sources

In the Lebas dataset [22], the authors explored the effects on the transcriptomic profile of *F. prausnitzii* A2-165 of treatment with supernatant from *Lactococcus lactis* subsp. lactis CNCM I-1631, *Lactococcus lactis* subsp. *cremoris* CNCM I-3558, *Lactobacillus paracasei* CNCM I-3689, and *Streptococcus thermophilus* CNCM I-3862. The raw RNA-seq fastq files are available at ArrayExpress under project E-MTAB-9387. In the D'hoe dataset, the authors sequenced RNA from *F. prausnitzii* A2-165 monocultures at three different time points [21]. The RNA-seq results were deposited in the Short Read Archive under the study identifier SRP136465. In the Kang dataset [23], the authors compared the transcriptomic profiles of *F. prausnitzii* A2-165 cultured using different carbohydrates (galactose, fructose, glucose, N-acetylglucosamine) as the sole carbon source. Data are in the Short Read Archive under the study identifier SRX10245665. Datasets and source code have been deposited in the GitLab repository at https://gitlab.com/adlin-science-public/FaePrau.

### Data processing

The workflow for this study is presented in S1 Fig. A read count table was obtained using the raw sequencing reads. After trimming reads with fastp v.0.20.0 (default parameters, [34]), fastq-formatted reads were aligned to the genome of *F. prausnitzii* strain A2-165 (Genome Assembly ASM273414v1) using BWA v.0.7.17 [35], allowing a single mismatch in the read. Then, sam-formatted alignments were sorted and converted to bam output files using SAMtools v.1.10 [36]. The number of reads per transcript from each sample was counted using HTSeqCount v.0.12.4 [37] and GFF-formatted gene annotations downloaded from NCBI. We checked the distribution of raw counts and performed principal component analysis in each dataset (S2 Fig). Gene expression values were normalized using the DEseq2 package v.1.34.0, in R [38]. The count table, containing 2,950 genes (S1 Table), was filtered to eliminate non-expressed genes. The resulting final dataset contained 2,902 genes and was further processed with WGCNA [39]. The quality of the expression matrix was evaluated by hierarchical clustering based on the distance between different samples, measured using Spearman's correlation. No outliers were detected (S3 Fig).

### Data analysis

Differential expression analysis was performed with the DEseq2 package v.1.34.0. We applied cutoffs of P-adj (corrected FDR) < 0.05 and absolute log2FoldChange (log2FC) > 1.5 for the detection of genes that were differentially expressed between the conditions considered and the negative control. Partial least squares-discriminant analysis (PLS-DA) implemented in the mixOmics package of R (v.6.18.1) [40] was used.

### Construction of weighted gene co-expression network

WGCNA is widely used in systems biology to construct a scale-free network from gene expression data [41]. In this study, we used the WGCNA package in R (v.1.71–3). In brief, Pearson's

correlation matrices were calculated for all pairs of genes and were subsequently transformed into matrices of connection strengths using a power function; in this case, the power was β = 17. We selected pairs of genes with distance correlation coefficients greater than 0.5 from the three datasets. The connection strengths were then used to calculate topology overlap (TO) [42], which measures the connectivity of a pair of genes. Hierarchical average linkage clustering [43] based on TO was used to identify gene co-expression modules, which groups genes with similar patterns of expression. We chose the minimum module size of 20 without merging the modules (a height cut of 0.0).

### Functional analysis and networks

After extracting the list of genes present in each module, we used eggNOG-mapper, a tool for functional annotation based on precomputed orthology assignments [44]. Functional enrichment analysis was performed in each module with STRING (version 9.0; https://string-db.org/) in March 2022. Enrichments with strength > 0.5 were selected. The resulting interaction networks contained direct (physical) and indirect (functional) interactions, derived from numerous sources including experimental repositories and computational prediction methods.

Nodes with a high degree of connectivity are called "hubs"; they interact with several other genes and may thus play a central role in the network. Instead, "betweenness" measures the total number of shortest nonredundant paths passing through a given node or edge. By definition, most of the shortest paths in a network go through nodes with a high degree of betweenness. These nodes, named "bottlenecks", become central points controlling the communication among other nodes in the network. In this study, we looked at the measures of connectivity and betweeness within each module by using a relative hub/bottleneck threshold of top 10%. Network visualization was performed with the igrah package v1.2.8.

## Results and discussion

### Comprehensive transcriptomic profiles from RNAseq data

Raw data from previous gene expression profiling analyse [21–23] were preprocessed using a similar workflow (S1 Fig). Details on these experiments and the samples used are presented in Table 1. To investigate differences in the transcriptomic profiles among the different conditions analyzed, we conducted a PLS-DA. This revealed a clear separation along the gene expression datasets from the different publications along the first two principal components, which explained 40% and 9% of the observed covariance (Fig 1A). While component 1 was sufficient to distinguish the Lebas dataset, the second component was necessary for separation of the D'hoe and Kang datasets. In total, we identified 50 genes that could be used to discriminate between the Lebas and Kang samples (component 1) and 50 genes that could be used to distinguish the D'hoe and Kang datasets (component 2) (Fig 1B). As shown in Fig 1C, the first component was principally influenced by clusters of orthologous genes (COGs) related to translation, carbohydrate transport and metabolism, cell cycle control, and lipid transport, and metabolism. The second component was more strongly affected by genes involved in coenzyme transport and metabolism, and inorganic ion transport and metabolism. In particular, among these discriminatory genes, we detected rubrerythrin (CG447_11045) and reverse rubrerythrin (CG447_01540), a putative iron transporter (CG447_11790 and CG447_11800), and enzymes of the cobalamin biosynthesis pathway (CG447_11805, CG447_11815, and CG447_11845). Rubrerythrin and reverse rubrerythrin are known to function primarily in the defense against oxidative radicals in anaerobic bacteria, while oxygen sensing and tight regulation of iron homeostasis are closely linked to bacterial growth in anaerobic conditions [45,46]. In addition, cobalamin has been demonstrated to have a protective effect against oxidative

**Table 1. Description of the dataset and samples used in this study.**

| Sample ID | Characteristics according to previous studies | Growth medium |
|---|---|---|
| ERR4363141 | control | Yeast extract-casein hydrolysate-fatty acids modified medium (YCFAm) containing 2 g/L cellobiose and 2 g/L glucose as carbon sources. Supernatants from several lactic acid bacteria/bifidobacteria strains were tested for their impact on growth of *F. prausnitzii*. [22] |
| ERR4363142 | control | |
| ERR4363143 | control | |
| ERR4363144 | *Lactobacillus paracasei* cell free supernatant | |
| ERR4363145 | *Lactobacillus paracasei* cell free supernatant | |
| ERR4363146 | *Lactobacillus paracasei* cell free supernatant | |
| ERR4363147 | *Lc. lactis* supernatant | |
| ERR4363148 | *Lc. lactis* supernatant | |
| ERR4363149 | *Lc. lactis* supernatant | |
| ERR4363150 | *Lactococcus lactis* subsp. *cremoris* cell free supernatant | |
| ERR4363151 | *Lactococcus lactis* subsp. *cremoris* cell free supernatant | |
| ERR4363152 | *Lactococcus lactis* subsp. *cremoris* cell free supernatant | |
| ERR4363153 | *Streptococcus thermophilus* cell free supernatant | |
| ERR4363154 | *Streptococcus thermophilus* cell free supernatant | |
| ERR4363155 | *Streptococcus thermophilus* cell free supernatant | |
| SRR6898059 | FP_3h | Medium for colon bacteria (mMCB) modified and containing 50 mM D-fructose as carbon source. [21] |
| SRR6898064 | FP_3h | |
| SRR6898055 | FP_9h | |
| SRR6898063 | FP_9h | |
| SRR6898054 | FP_15h | |
| SRR6898060 | FP_15h | |
| SRR13865118 | galactose | Yeast extract-casein hydrolysate-fatty acids (YCFA) medium supplemented with 0.5 w/v carbon sources, vitamin solution and short-chain fatty acids (v/v). [23] |
| SRR13865119 | galactose | |
| SRR13865120 | glucose | |
| SRR13865121 | glucose | |
| SRR13865122 | NAG | |
| SRR13865123 | NAG | |
| SRR13865124 | fructose | |
| SRR13865125 | fructose | |

*NAG: *N*-acetylglucosamine.

stress in some bacteria [47]. Therefore, it appears that some of the key discriminatory genes we identified are functionally related to each other and associated with mechanisms for protection against oxidative stress, which could be essential for the growth of *F. prausnitzii* under the different anaerobic conditions investigated in the three previous publications.

## Exploration of transcriptomic data with R-Shiny

We developed an R-Shiny application to facilitate deeper exploration of the transcriptomic data from each of the three studies. For example, using this approach on the Lebas dataset, we observed that treatment of *F. prausnitzii* with supernatant from *L. lactis* subsp. *lactis* generated more transcriptional downregulation than upregulation relative to the control (see our R-Shiny application), confirming the findings of the original study [22]. In the Kang dataset, a comparison between bacteria raised on glucose or fructose as the sole carbon source highlighted that the most upregulated gene was CG447_08360, which encodes a putative PTS glucose transporter subunit IIBC (Fig 2A and 2B). This gene was also highly upregulated in the

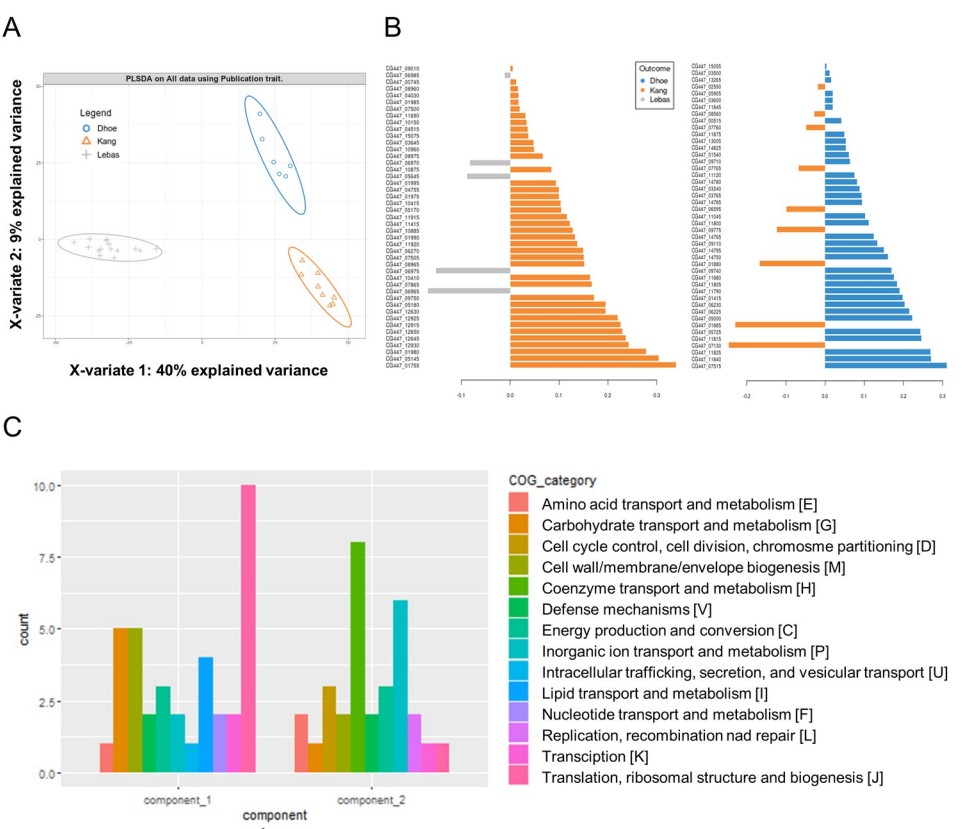

**Fig 1. Partial least-square discriminant analysis of RNA-seq data.** (A) PLS-DA plot of RNA-seq data showing clear transcriptome-based discrimination among the three publication datasets. Each point represents the transcriptome signature of one sample, with ellipses representing 95% confidence level. (B) Genes with high discriminatory ability were identified from PLS-DA. (C) Histogram showing the distribution of COG categories (Clusters of Orthologous Genes) associated with the genes with the highest discriminatory power in the two components generated by PLS-DA. See S2 Table for the full list of discriminatory genes.

presence of galactose. As the transcription PTS genes is usually regulated in response to substrate availability [48], this suggests that CG447_08360 encodes a galactose/glucose transporter. Finally, the D'hoe dataset provided the opportunity to track *F. prausnitzii* gene expression across three growth time-points. In the R-Shiny application, it was possible to compare, for example, expression at time 15h with that at time 3h and to highlight the most upregulated and downregulated genes between these time-points (Fig 2C–2H). Notably, serine/threonine and ferrous iron transporters were highly upregulated at the earlier time compared to later, which suggests that available serine, threonine, and ferrous iron are imported into the cell during early growth (Fig 2D and 2E). It was also interesting to observe the upregulation of CG447_14545 at time 3h (Fig 2F); this gene encodes a multi-antimicrobial extrusion (MATE) efflux transporter, which plays a role in antibiotic resistance as well as potentially in bacterial-host interactions and intercellular signaling [49,50]. Instead, time-point 15h was marked by strong upregulation of CG447_09920 and CG447_06950, which encode energy-coupling factor (ECF) transport S component and DNA polymerase IV, respectively (Fig 2G and 2H). ECF transporters are responsible for vitamin uptake in prokaryotes [51], while DNA polymerase IV is part of the SOS response to DNA damage in bacteria and contributes significantly to cell fitness in late stationary phase cultures in the absence of any exogenous DNA damage [52]. Our

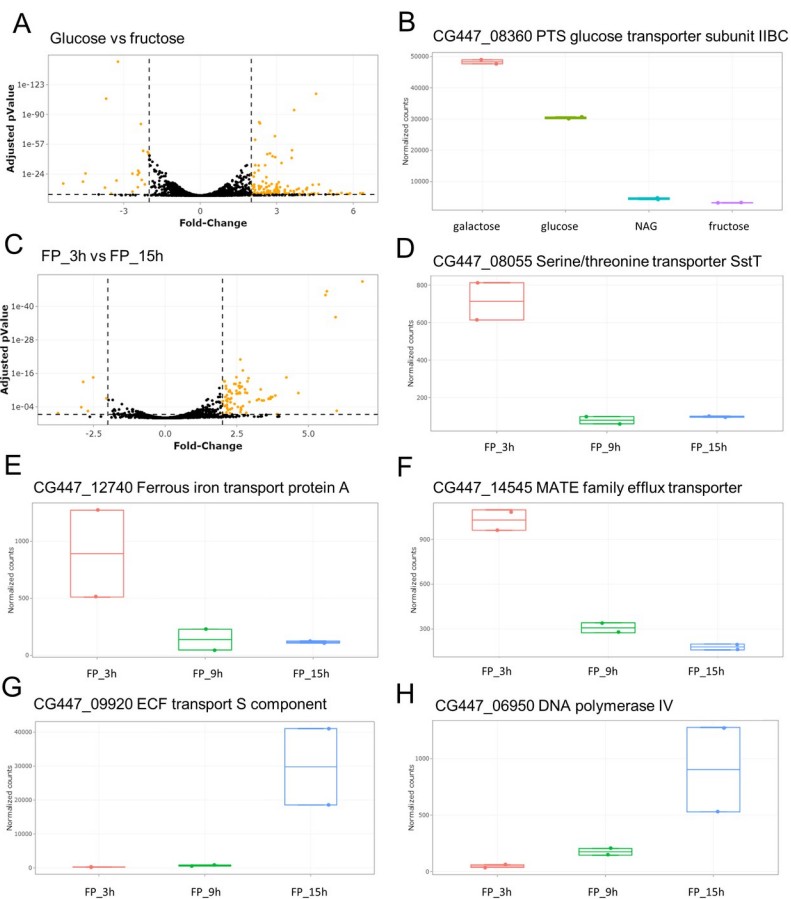

**Fig 2. Illustration of transcriptomic data mining.** (A) and (C) Visualization of RNAseq results with volcano plots. (B, D, E, F, G, H) Boxplots showing the read counts for genes of interest among different growth conditions.

R-Shiny application thus provided new insight into each dataset and reveals genes that may play a role in the fitness of *F. prausnitzii* under the different growth conditions employed.

## Construction of weighted gene co-expression network

To choose the best soft-thresholding power, we analyzed network topology and determined that 17 was the lowest appropriate power value, with a scale-free topology fit index of 0.95 and a relatively high average connectivity (Fig 3A). WGCNA revealed 24 distinct modules representing highly co-expressed networks of genes (Fig 3B–3D). The list of genes in each module of the WGCNA analysis is available in S3 Table. Two modules, turquoise and blue, were more notably larger than the rest, with 568 and 431 genes, respectively (Table 2). The grey module contained the non-clustering genes, of which there were only 36, suggesting that WGCNA performed well under the parameters applied here.

Individual modules can represent independent units that are responsible for certain biological functions. By linking the gene expression of the modules with the experimental conditions employed, it may be possible to identify modules that have important functions under certain conditions. To this end, we used WGCNA to correlate each module with the conditions used in the three publications (Lebas, D'hoe and Kang) by calculating values of Module Significance (MS) for module-trait correlations (Fig 4A). The expression levels for each module across the

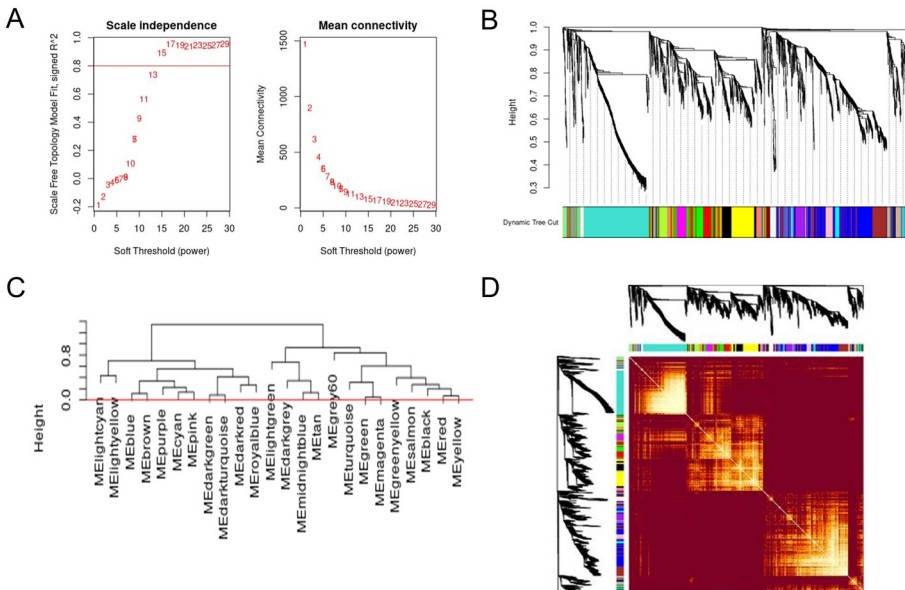

**Fig 3. Network construction and module detection with WGCNA.** (A) Network topology analysis for various values of soft-thresholding powers, with the scale-free index and the mean connectivity as a function of the soft-thresholding power. (B) Dendrogram of all genes divided into 24 modules, with dissimilarity based on topological overlap, presented with assigned module colors. (C) Dendogram representing the 24 modules identified by WGCNA. The heightcut (red line, heightcut = 0.0) was used to unmerge modules. The grey module represents genes that are not included in any of the other modules. (D) A heatmap depicting the topological overlap matrix (TOM) among all genes in the analysis. The intensity of the red color indicates the strength of the correlation between all pairs of genes.

24 samples were then visualized (Fig 4B). Several large modules represent expression changes that clearly delineate between Lebas on one side and D'hoe and Kang on the other. Indeed, the blue, cyan, pink, brown, and purple modules (Pearson correlation coefficient ≥ 0.75) correspond to genes that were upregulated in the D'hoe and Kang datasets, while the green, salmon, red, and yellow modules (Pearson correlation coefficient ≤ - 0.75) represent genes that were upregulated in the Lebas dataset. This suggests that the expression patterns of the genes in these modules reflect the ways in which the growth of *F. prausnitzii* differed among the conditions used in the different studies.

## Functional and pathway enrichment analysis

The top biological processes associated with the large brown and blue modules were "structural constituent of ribosome" and "regulation of translation", respectively (Table 3). Several modules were enriched in metabolic functions such as arginine, histidine, cobalamin, or fatty acid biosynthesis (green, red, lightgreen, or royal blue modules, respectively; Table 3 and Fig 5). We also identified clusters of genes related to bacteriophage function, chaperones, stress response, or SOS response, suggesting the involvement of functions related to adaptive responses (darkred, tan and black modules). Overall, we found that genes involved in similar pathways or with the same biological function tended to belong to the same expression cluster with WGCNA, which supports the relevance and utility of this integrative bioinformatics approach in this context.

In their original study, Lebas et al. showed that *F. prausnitzii* A2-165 responds to cell-free supernatants from lactic acid bacteria by downregulating mobilome genes and upregulating cell-wall related genes. Here, we observed that the purple module (downregulated in Lebas relative to D'hoe and Kang) contained clusters of genes related to transposon and mobilization

**Table 2. Size of gene co-expression modules.**

| Module color | Number of genes |
|---|---|
| black | 144 |
| blue | 431 |
| brown | 270 |
| cyan | 64 |
| darkgreen | 28 |
| darkgrey | 21 |
| darkred | 30 |
| darktruquoise | 28 |
| green | 175 |
| greenyellow | 83 |
| grey | 36 |
| lightcyan | 50 |
| lightgreen | 34 |
| lightyellow | 33 |
| magenta | 85 |
| midnightblue | 54 |
| pink | 89 |
| purple | 83 |
| red | 164 |
| royalblue | 31 |
| salmon | 71 |
| tan | 76 |
| turquoise | 568 |
| yellow | 254 |

proteins, while conversely, the yellow module (upregulated in Lebas relative to D'hoe and Kang) was linked with processes associated to the cell wall and the regulation of cell shape. In addition, we found that some genes involved in mobilization or cell-wall synthesis form potential networks of interaction (Table 3). In this way, the results from our WGCNA strengthen the observations by Lebas et al.

## Identification of hub and bottleneck genes

Next, we focused on the genes with the highest intramodular connectivity (hub genes), as these serve as the best representatives of the expression of the module as a whole [39] and play important roles in biological processes. Network analysis highlighted the 25 genes with highest connectivity, and all belong to the turquoise module (Fig 6 and Table 4). Genes with the highest degree of betweenness (called bottlenecks) control most of the information flow, and thus represent the critical points of the network [53]. In our network, we observed that the top 25 bottleneck genes also belonged to the turquoise module, and they tended to be hubs as well (Table 4). Among these, we identified genes encoding proteins involved in DNA metabolism such as DNA topoisomerase III (CG447_13500 and CG447_13565), DNA primase (CG447_13520 and CG447_13625), relaxase (CG447_02250), and integrase (CG447_13030), as well as group II intron reverse transcriptase/maturase (CG447_11550). When we analyzed the location of these genes on the chromosome, we found that they were distributed in seven regions: six encoded mobile genetic elements (MGEs) and one carried prophage genes. Interestingly, the TraG protein family, encoded by putative MGEs 1, 4, and 6 (Table 4), has been

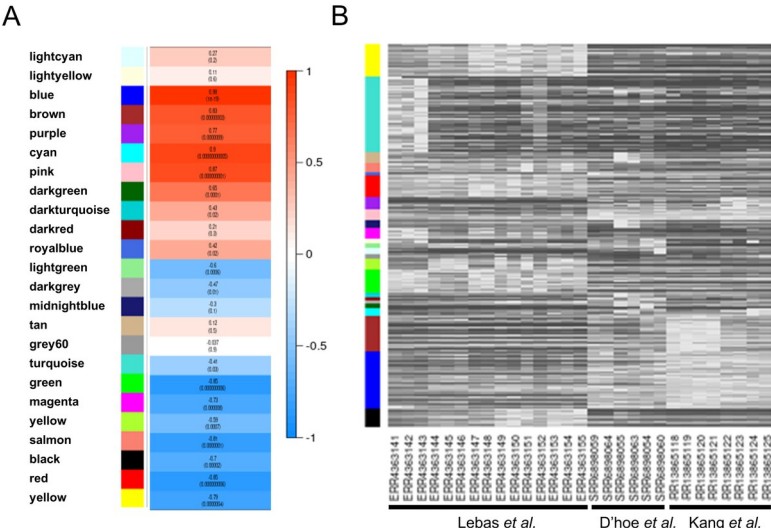

**Fig 4. Identification of the modules associated with conditions in the three original datasets.** (A) Heatmap depicting the correlation between module eigengenes and the original datasets. Pearson coefficient correlations are indicated. The *p*-value is indicated in parentheses. (B) Heatmap of gene expression levels in the modules across the samples in the three original datasets.

shown to be involved in bacterial conjugation [54]. However, to our knowledge, no conjugation system has been described in *F. prausnitzii* and no endogenous plasmid was isolated in the sequenced strains. The absence of conjugative plasmids but the presence of many conjugation-related genes suggests the presence of integrative and conjugative elements (ICEs) in *F. prausnitzii*. In addition, we observed that the six MGEs highlighted in the turquoise module were strongly associated within a gene network. This raises questions about possible shared means of regulation and/or induction for these elements.

The conditions in the gut, which are characterized by extremely high concentrations of microbial cells and phages, represent one of the most favorable ecological niches for horizontal gene exchange. Indeed, the gut microbiome responds to selective pressure through the genetic restructuring of gut populations, driven mainly via horizontal gene exchange [55,56]. Recently, it was shown that ICEs play an important role in the adaptation of *Streptococcus salivarius* to human oral, pharyngeal and gut environments [57]. Indeed, ICEs encode numerous functions such as resistance to stress or antibiotics and numerous enzymes involved in diverse cellular metabolic pathways. Further investigation is needed to understand the role(s) played by MGEs in the evolution of the *F. prausnitzii* genome, their influence on the dynamic response to selective pressure (i.e. antibiotics, IBDs), and the consequences for host health.

## Conclusions

Despite the importance of *F. prausnitzii* to gut health, the gene regulatory networks that determine the behavior of this bacterium within its ecological niche are still unknown. Using an integrative bioinformatics approach, this work describes the first reconstruction of co-expression networks in *F. prausnitzii* strain A2-165. Our findings provide insights into the relationships among several cellular processes, and provide evidence for a tight network of mobile genetic elements. This work highlights candidate genes that should be further investigated for their role(s) in the response of *F. prausnitzii* to different factors in the gut, including antibiotics, therapies, lifestyle, or diet. As *F. prausnitzii* is considered a promising next generation

**Table 3. Functional enrichment in each module, analyzed with STRING 9.0.**

| Module name | Term description | Observed gene count | Background gene count | Strength |
|---|---|---|---|---|
| black | Catalytic complex | 9 | 52 | 0.64 |
| | Incl. Glycosyl hydrolase-like 10 | 5 | 7 | 1.25 |
| | SOS response | 5 | 8 | 1.2 |
| blue | Oxidoreductase activity, acting on NAD(P)H | 6 | 7 | 0.85 |
| | Regulation of translation | 9 | 18 | 0.62 |
| | Electron transport chain | 11 | 24 | 0.58 |
| | tRNA aminoacylation for protein translation | 11 | 25 | 0.56 |
| | Polysaccharide biosynthetic process | 12 | 28 | 0.55 |
| | Pyrimidine-containing compound metabolic process | 16 | 40 | 0.52 |
| | Polysaccharide metabolic process | 15 | 38 | 0.52 |
| brown | Structural constituent of ribosome | 33 | 52 | 0.93 |
| | rRNA binding | 25 | 42 | 0.9 |
| | RNA binding | 45 | 133 | 0.65 |
| | tRNA binding | 10 | 30 | 0.65 |
| cyan | Proton transmembrane transporter activity | 5 | 20 | 1.18 |
| dargreen | Incl. side of membrane, cdar, ggdef-like domain | 3 | 5 | 1.89 |
| | Incl. hypoxanthine catabolic process, and selenocompound metabolism | 5 | 12 | 1.73 |
| darkred | Incl. bacteriophage mu, gpt, and gp36 | 9 | 14 | 1.9 |
| | Incl. peptidyl-lysine methylation | 6 | 12 | 1.79 |
| | Incl. transposition and vancomycin resistance | 12 | 22 | 1.83 |
| green | Arginine biosynthesis | 5 | 6 | 1.24 |
| | Incl.d l-2-amino-thiazoline-4-carboxylic acid hydrolase | 7 | 14 | 1.02 |
| greenyellow | Ribosome biogenesis | 7 | 43 | 0.85 |
| lightcyan | *De novo* nucleotide biosynthetic process | 9 | 12 | 1.78 |
| lightgreen | Cobalamin biosynthetic process | 16 | 21 | 1.92 |
| lightyellow | Incl. dna alkylation and sulfate reduction | 20 | 23 | 1.98 |
| | Incl. irre n-terminal-like domain and flavodoxin | 4 | 14 | 1.49 |
| magenta | Incl. cysteine protease prp and terminase-like family | 9 | 11 | 1.53 |
| | Incl. feoa domain and cysteine transport | 4 | 8 | 1.32 |
| midnightblue | Amino sugar catabolic process and hexokinase activity | 5 | 5 | 1.83 |
| purple | Molybdenum cofactor biosynthesis | 6 | 9 | 1.48 |
| | Incl. transposon-encoded protein tnpw, and ecf sigma factor | 9 | 11 | 1.57 |
| | Incl. endonuclease relaxase, moba/vird2, and bacterial mobilization protein (mobc) | 4 | 5 | 1.56 |
| red | Histidine biosynthetic process | 6 | 10 | 1.11 |
| royalblue | Fatty acid biosynthetic process | 8 | 15 | 1.79 |
| | Response to toxic substance | 3 | 14 | 1.39 |
| salmon | Incl. zinc-ribbon domain and RNA polymerase sigma-70 like | 3 | 5 | 1.54 |
| | Incl. brxa and pglz domain | 5 | 20 | 1.16 |
| tan | Stress response | 5 | 8 | 1.47 |
| | Chaperone | 6 | 15 | 1.28 |
| turquoise | Incl. tail tube protein, and c-type lectin fold | 59 | 72 | 0.74 |
| | Incl. aaa-like domain, and yodl-like | 51 | 80 | 0.63 |
| | Incl. maff2 family, and nlpc/p60 family | 35 | 56 | 0.63 |
| yellow | Cell wall | 9 | 28 | 0.67 |
| | Regulation of cell shape | 11 | 23 | 0.85 |

Enrichments with strength > 0.5 were selected. Incl. means Include.

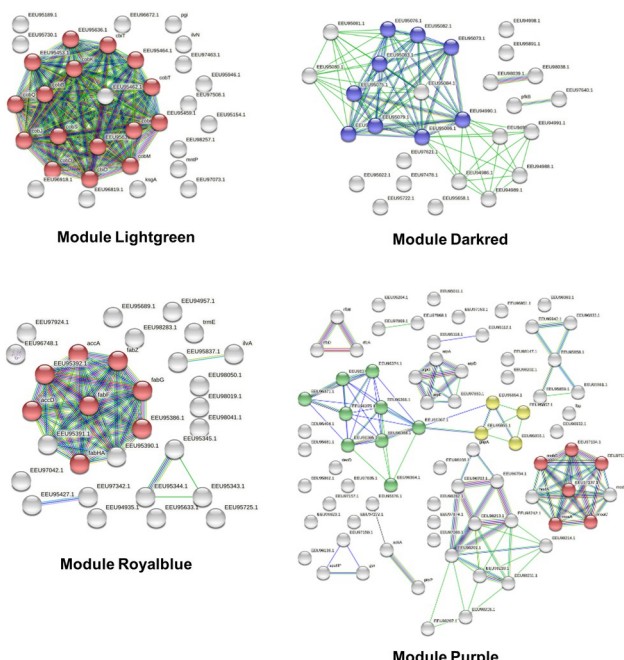

**Fig 5. Illustration of enrichment analysis with STRING (version 9.0).** Results are presented for the modules "Lightgreen" (red color: Cobalamin biosynthetic pathway), "Darkred" (blue color: Bacteriophage functions), "Royalblue" (red color: Fatty acid biosynthesis), "Purple" (green color: Transposon-encoded proteins; red color: Molybdenum cofactor biosynthesis; yellow color: Endonuclease/relaxase).

probiotic, our results raise the question of its functionality, which seems related to environmental conditions. This aspect must be taken into account for the optimization in the administration of *F. prausnitzii* as a probiotic in patients with an altered digestive ecosystem. In addition, we provide an online tool at https://faeprau.omics.ovh/ for data exploration through a user-friendly graphical interface that may be useful for the scientific community. In the

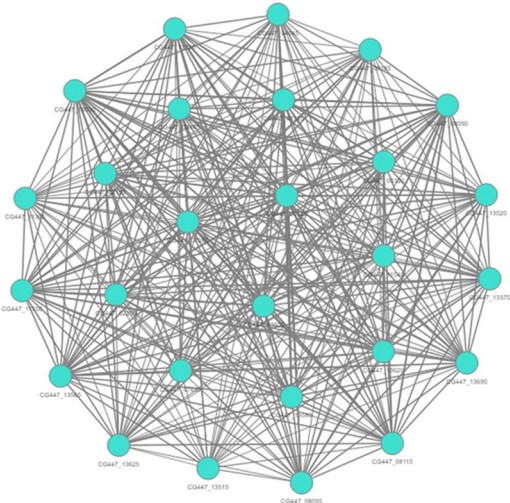

**Fig 6. Network plot depicting the top connections in the "turquoise" module.** Nodes represent genes, and node size is correlated with the degree of connectivity of the gene.

**Table 4. Identification of hub and bottleneck genes from the WGCNA network.**

| Gene | Deg | Betweenness | Betweenness2 | Hub score | Putative function |
|------|-----|-------------|--------------|-----------|-------------------|
| **Putative mobile element 1** | | | | | |
| CG447_00320 | 220 | 409.41 | 50 | 0.979 | chromosome partitioning protein ParB |
| CG447_00340 | 231 | 702.79 | 342 | 0.991 | conjugal transfer protein TraG |
| **Putative mobile element 2** | | | | | |
| CG447_02190 | 221 | 414.15 | 17 | 0.983 | helix-turn-helix domain-containing protein |
| CG447_02250 | 226 | 555.10 | 699 | 0.985 | relaxase |
| **Putative mobile element 3** | | | | | |
| CG447_07080 | 229 | 524.75 | 41 | 0.990 | protein BART-1 |
| CG447_07630 | 222 | 551.30 | 179 | 0.969 | hypothetical protein |
| **Putative mobile element 4** | | | | | |
| CG447_08090 | 236 | 793.09 | 40 | 0.997 | conjugal transfer protein TraG |
| CG447_08095 | 215 | 338.09 | 115 | 0.975 | condensin complex subunit 2 |
| CG447_08115* | 211 | 311.56 | 7 | 0.968 | conjugal transfer protein TraE |
| CG447_08125 | 230 | 661.10 | 469 | 0.990 | CHAP domain-containing protein |
| CG447_08145 | 220 | 393.73 | 54 | 0.982 | recombinase |
| CG447_08185 | 214 | 409.66 | 221 | 0.960 | hypothetical protein |
| **Putative mobile element 5** | | | | | |
| CG447_11365 | 217 | 482.68 | 330 | 0.960 | hypothetical protein |
| CG447_11540 | 214 | 460.67 | 1275 | 0.951 | PrgI family protein |
| CG447_11550 | 245 | 1242.96 | 271 | 1.000 | group II intron reverse transcriptase/maturase |
| **Putative mobile element 6** | | | | | |
| CG447_13030 | 237 | 786.67 | 58 | 0.997 | site-specific integrase |
| CG447_13500 | 244 | 1223.08 | 252 | 0.998 | DNA topoisomerase III |
| CG447_13515* | 177 | 308.72 | 1178 | 0.806 | virulence-associated protein E |
| CG447_13520 | 225 | 651.28 | 422 | 0.971 | DNA primase |
| CG447_13565 | 233 | 752.36 | 134 | 0.987 | DNA topoisomerase |
| CG447_13570 | 230 | 538.49 | 75 | 0.992 | DUF4366 domain-containing protein |
| CG447_13580 | 212 | 317.01 | 10 | 0.970 | peptidase M23 |
| CG447_13620 | 213 | 324.64 | 13 | 0.972 | virulence-associated protein E |
| CG447_13625 | 212 | 355.56 | 263 | 0.964 | DNA primase |
| CG447_13690 | 225 | 484.23 | 259 | 0.986 | conjugal transfer protein TraG |
| CG447_13725 | 220 | 437.92 | 246 | 0.979 | recombinase |
| **Putative phage region** | | | | | |
| CG447_04045 | 220 | 447.01 | 178 | 0.975 | IS30 family transposase |

All genes belong to "turquoise" module. Asterisks indicate hub genes. All others are hub-bottleneck genes.

future, this interface may evolve in order to add RNAseq data generated by research teams studying *F. prausnitzii*.

## Supporting information

**S1 Fig. Flow chart of data preparation, processing, and analysis.**
(TIF)

**S2 Fig. Exploratory analyses of each dataset.** After processing the raw data, we checked the distribution of raw counts and performed principal component analysis.
(TIF)

**S3 Fig. Sample clustering.** The quality of the expression matrix was evaluated by sample clustering based on the distance between different samples, measured as Spearman's correlation. No outliers were detected.
(TIF)

**S1 Table. Count table from RNAseq data.**
(XLSX)

**S2 Table. List of discriminatory genes in components 1 and 2 with PLS-DA analysis.** The tool eggNOG-mapper was used for functional annotation based on precomputed orthology assignments.
(XLSX)

**S3 Table. List of the genes in the detected modules.**
(XLSX)

## Author Contributions

**Conceptualization:** Sandrine Auger, Virginie Mournetas, Hélène Chiapello, Valentin Loux.

**Formal analysis:** Sandrine Auger, Virginie Mournetas.

**Supervision:** Sandrine Auger.

**Visualization:** Sandrine Auger, Virginie Mournetas.

**Writing – original draft:** Sandrine Auger, Virginie Mournetas.

**Writing – review & editing:** Philippe Langella, Jean-Marc Chatel.

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
