## [Decision Letter · Decision Letter 0]

30 Aug 2022

PONE-D-22-19206Gene co-expression network analysis of the human gut commensal bacterium Faecalibacterium prausnitzii based on WGCNA in R-ShinyPLOS ONE

Dear Dr. Auger,

Thank you for submitting your manuscript to PLOS ONE. After careful consideration, we feel that it has merit but does not fully meet PLOS ONE’s publication criteria as it currently stands. Therefore, we invite you to submit a revised version of the manuscript that addresses the points raised during the review process.

Please address the concerns noted by the reviewer. Please submit your revised manuscript by Oct 14 2022 11:59PM. If you will need more time than this to complete your revisions, please reply to this message or contact the journal office at plosone@plos.org. Please include the following items when submitting your revised manuscript:A rebuttal letter that responds to each point raised by the academic editor and reviewer(s). You should upload this letter as a separate file labeled 'Response to Reviewers'.A marked-up copy of your manuscript that highlights changes made to the original version. You should upload this as a separate file labeled 'Revised Manuscript with Track Changes'.An unmarked version of your revised paper without tracked changes. You should upload this as a separate file labeled 'Manuscript'.If applicable, we recommend that you deposit your laboratory protocols in protocols.io to enhance the reproducibility of your results. Protocols.io assigns your protocol its own identifier (DOI) so that it can be cited independently in the future. For instructions see: https://journals.plos.org/plosone/s/submission-guidelines#loc-laboratory-protocols. Additionally, PLOS ONE offers an option for publishing peer-reviewed Lab Protocol articles, which describe protocols hosted on protocols.io. Read more information on sharing protocols at https://plos.org/protocols?utm_medium=editorial-email&utm_source=authorletters&utm_campaign=protocols.

We look forward to receiving your revised manuscript.

Kind regards,

Brenda A Wilson, Ph.D.

Academic Editor

PLOS ONE

Journal Requirements:

Additional Editor Comments:

Both reviewers felt the manuscript was interesting and informative. One reviewer noted some issues that should be addressed. I agree with this assessment.

Reviewers' comments:

Reviewer's Responses to Questions

**Comments to the Author**

1. Is the manuscript technically sound, and do the data support the conclusions?

Reviewer #1: Yes

Reviewer #2: Yes

2. Has the statistical analysis been performed appropriately and rigorously? 

Reviewer #1: Yes

Reviewer #2: Yes

3. Have the authors made all data underlying the findings in their manuscript fully available?

Reviewer #1: Yes

Reviewer #2: Yes

4. Is the manuscript presented in an intelligible fashion and written in standard English?

Reviewer #1: Yes

Reviewer #2: Yes

5. Review Comments to the Author

Reviewer #1: Authors analyze mRNA datasets from F. prausnitzii using the strain A2-165 as a reference and use an algorithm to show co-expression patterns and postulate expression networks in an integrative bioinformatics approach. The work consists of a further analysis based on gene regulation and the physiological complexity of organisms by looking for patterns of co-expressions. The paper makes a networked approach for gene expression in Fpra instead of particular one-gene-linked activities.

Further, the work links hubs and bottlenecks to specific physiological situations under which the mRNA dataset was obtained. These datasets (Lebas, D'hoe, and Kang) are well differentiated using PCA, probably because these are a result of different experimental conditions. This work goes beyond connecting all gene expressions through a network that outlines the importance of functional modules.

This analysis has already been performed on other organisms, but it takes special relevance with this particular species since it is considered one of the most relevant indicators of healthy intestinal microbiota.

In the introduction, the authors mention the idea of using F. prausnitzii as a "real" probiotic. However, the results of this work indicate that Faecalibacterium prausnitzii alone might not be sufficient in case it could be uptaken as a probiotic since its functionality seems to be highly dependent on environmental conditions. If this is true, it would be interesting to add a comment regarding this point of view, considering that this probiotic would be given to patients with sometimes heavily altered ecosystems.

The paper is well structured and well written. No significant issues have been found in the manuscript. Some questions and specific comments follow:

The provided website (https://feaprau.omics.ovh/) for data exploration does not work. Will it be available after the publication of the paper?

¿Is there any relationship between the outcome of the analysis and the composition of the culture media used?

Line 209: "are to known to function"; delete first "to"

Line 276: "have important functions in certain conditions.", consider "...under certain conditions".

Reviewer #2: The manuscript by Dr. Auger and colleagues presents a systems biology approach where data of expression in RNA seq datasets from F. prausnitzii A2-165 has been analysed. The article is well written and the study is undoubtedly scientifically sound, because limited information exists to date concerning gene expression and regulation for this microorganism, despite its association for human well-being.

However, I have some minor points to be addressed by the authors:

1. Consider in the title to develop WGCNA acronym or rephrase to avoid its use.

2. L92: MAM acronym should be placed after its definition.

3. In general, figures were difficult to see clearly. Please, consider to upload an improved version for all of them (main figures and those in supplementary material) with better resolution.

4. Additional points to be discussed/comment further:

• It is not clear to me if in the future, it will be possible to include in the tool developed additional data of RNA seq studies that will arise from the same strain in other conditions or from other F. prausnitzii strains. Please, discuss further on this possibility.

• L195-197 the authors evidence that the gene expression datasets cluster apart. Table 1 evidences that a different number of samples (and therefore reads) have been included in the analysis for each study. However, all are analysed together in the WGCNA construction. I wonder if there could be a bias due to this differences in number of samples/reads. How could this have influenced in the weighted gene co-expression network? (for example that the network modules are more representative of Lebas study). If the authors have performed any test to control this is not taking place, please provide further details.

• In the identification of hub and bottleneck genes a threshold value has been used (>210 degree of connectivity and >300 betweenness, respectively). As far as I understood, no correction per number of genes in the module has been carried out. I suggest to add an analysis considering this additional parameter because in modules with a limited number of genes connectivity among them could be also high, but not reaching the thresholds set simply because they are smaller modules. Then the % of connectivity/betweenness considering the number of genes in the module could be evaluated to identify genes of interest.

For this reviewer it has not been possible to fully evaluate the utility of the tool as it was not publicly available. I suggest for future submissions to consider implementing an option of access for those who have to review the work.

6. PLOS authors have the option to publish the peer review history of their article (what does this mean?). If published, this will include your full peer review and any attached files.

Reviewer #1: **Yes: **L. Jesús Garcia-Gil

Reviewer #2: No

---

## [Author Response · Author response to Decision Letter 0]

10 Oct 2022

Reviewer #1

1. In the introduction, the authors mention the idea of using F. prausnitzii as a "real" probiotic. However, the results of this work indicate that Faecalibacterium prausnitzii alone might not be sufficient in case it could be uptaken as a probiotic since its functionality seems to be highly dependent on environmental conditions. If this is true, it would be interesting to add a comment regarding this point of view, considering that this probiotic would be given to patients with sometimes heavily altered ecosystems.

We agree with Reviewer�1 who raises a sensitive point on the use of F. prausnitzii as a probiotic and even on the use of probiotics in general. We added a comment in the Conclusion section:

“As F. prausnitzii is considered a promising next generation probiotic, our results raise the question of its functionality, which seems related to environmental conditions. This aspect must be taken into account for the optimization in the administration of F. prausnitzii as a probiotic in patients with an altered digestive ecosystem.”

2. The provided website (https://feaprau.omics.ovh/) for data exploration does not work. Will it be available after the publication of the paper?

We apologize for the inconvenience. Now the site is in public access. It is accessible at the following link: https://faeprau.omics.ovh/.

3. ¿Is there any relationship between the outcome of the analysis and the composition of the culture media used?

In the three studies, the media used were rich media whose composition was partially defined. They all contained broths of peptones, casa amino acids, or yeast extracts. They were supplemented with carbon sources, vitamins and/or short chain fatty acids. We tested if there was a relationship between the modules defined by WGCNA and the carbon source used in the different samples. In addition, we also tested whether there was a relationship between the modules and the supplementation of short-chain fatty acids in the medium. However, we did not find any relationship between the outcome of the analysis and the carbon sources or the short-chain fatty acids present in the culture media. 

We believe that more RNAseq data are necessary to highlight precise relationships between the outcomes and the composition of the culture medium. We hope to enrich and update our analyses with future available RNAseq data.

4. Line 209: "are to known to function"; delete first "to"

Done

5. Line 276: "have important functions in certain conditions.", consider "...under certain conditions".

Done

Reviewer #2

1. Consider in the title to develop WGCNA acronym or rephrase to avoid its use.

We reconsidered the title to avoid the use of the method WGCNA: “Gene co-expression network analysis of the human gut commensal bacterium Faecalibacterium prausnitzii in R-Shiny”.

2. L92: MAM acronym should be placed after its definition.

Done

3. In general, figures were difficult to see clearly. Please, consider to upload an improved version for all of them (main figures and those in supplementary material) with better resolution.

We have reviewed the resolution of the figures.

4. Additional points to be discussed/comment further:

• It is not clear to me if in the future, it will be possible to include in the tool developed additional data of RNA seq studies that will arise from the same strain in other conditions or from other F. prausnitzii strains. Please, discuss further on this possibility.

Indeed, we have developed this tool with the aim of making it evolve by adding future public RNAseq data from other research teams studying the A2-165 strain.

We are studying two possibilities: 

(1) the researchers will upload their data in a transitory way on the interface, but there is a risk of bias in normalizing the data.

(2) they will contact the corresponding authors in order to normalize on all the datasets, and so that their data is permanently added to the interface.

For now, we have focused on the strain A2-165 as it is the only strain for which there is RNAseq data. As suggested by Reviewer #2, we can consider integrating RNAseq data from other strains in the future.

We added a sentence at this end of the discussion section: “In the future, this interface may evolve in order to add RNAseq data generated by research teams studying F. prausnitzii.”

• L195-197 the authors evidence that the gene expression datasets cluster apart. Table 1 evidences that a different number of samples (and therefore reads) have been included in the analysis for each study. However, all are analysed together in the WGCNA construction. I wonder if there could be a bias due to this differences in number of samples/reads. How could this have influenced in the weighted gene co-expression network? (for example that the network modules are more representative of Lebas study). If the authors have performed any test to control this is not taking place, please provide further details.

WGCNA is an unsupervised method, which analyzes and aggregates unlabeled datasets. Thus, the difference in number of samples in each datasets does not influence the construction of the co-expression networks. As the number of reads is different from one sample to another, we applied a normalization step to take these differences into account.

In Figure 1A, the PLSDA showed that the Lebas dataset is strongly separated from the other two datasets. In addition, we also observed by principal component analysis, which ignores the information regarding the class of the samples, that the samples from the Lebas dataset are also strongly separated from the others. It is therefore not surprising that certain genes in the modules are only co-expressed in the Lebas dataset. 

The PCA can be viewed at the following link: https://faeprau.omics.ovh/.

• In the identification of hub and bottleneck genes a threshold value has been used (>210 degree of connectivity and >300 betweenness, respectively). As far as I understood, no correction per number of genes in the module has been carried out. I suggest to add an analysis considering this additional parameter because in modules with a limited number of genes connectivity among them could be also high, but not reaching the thresholds set simply because they are smaller modules. Then the % of connectivity/betweenness considering the number of genes in the module could be evaluated to identify genes of interest.

We thank Rewiever #2 for this very interesting remark. In gene co-expression network, hub genes (with high degree of connectivity) interact with many other genes. On other hand, the measure of betweenness is based on the theory that important genes lie on many paths between other genes in the network. In this context, we were interested in the genes with the highest values of connectivity and/or betwenness because they are likely to be biologically important for the pathways or processes represented by a particular module. 

We agree with Rewiever#2 that modules containing fewer genes could also have strongly connected genes. Thus, we looked at the measures of connectivity and betweeness within each module without applying an absolute hub/bottleneck threshold but by using a relative hub/bottleneck threshold of top 10%. As a result, we detected only one gene in the module lightyellow (12 genes in total) that could be considered as a hub and/or a bottleneck (with a degree of connectivity = 10 and a betweenness = 45). And we detected only 3 hub genes (degree of connectivity = 17) and 4 bottleneck genes (betweenness > 31) in the module grey (27 genes in total). These 7 genes are located in the same operon. Thus, we are not sure of the relevance of this observation. More RNAseq experiments may lead to more robust and refined results. We hope to enrich and update our analyses with future available RNAseq data.

For this reviewer it has not been possible to fully evaluate the utility of the tool as it was not publicly available. I suggest for future submissions to consider implementing an option of access for those who have to review the work.

We apologize for the inconvenience. Now the site is in public access. It is accessible at the following link: https://faeprau.omics.ovh/.

---

## [Editor Report · Decision Letter 1]

6 Nov 2022

Gene co-expression network analysis of the human gut commensal bacterium Faecalibacterium prausnitzii in R-Shiny

PONE-D-22-19206R1

Dear Dr. Auger,

We’re pleased to inform you that your manuscript has been judged scientifically suitable for publication and will be formally accepted for publication once it meets all outstanding technical requirements.

Kind regards,

Brenda A Wilson, Ph.D.

Academic Editor

PLOS ONE

---

## [Editor Report · Acceptance letter]

10 Nov 2022

PONE-D-22-19206R1 

Gene co-expression network analysis of the human gut commensal bacterium *Faecalibacterium prausnitzii* in R-Shiny 

Dear Dr. Auger:

I'm pleased to inform you that your manuscript has been deemed suitable for publication in PLOS ONE. Congratulations! Your manuscript is now with our production department. 

Kind regards, 

on behalf of

Dr. Brenda A Wilson 

Academic Editor

PLOS ONE